# SemiAugIR: Semi-supervised Infrared Small Target Detection via Thermodynamics-Inspired Data Augmentation

## Abstract

Convolutional neural networks have shown promising results in single-frame infrared small target detection (SIRST) through supervised learning. Nevertheless, this approach requires a substantial number of accurate manual annotations on a per-pixel basis, incurring significant labor costs. To mitigate this, we pioneer the integration of semi-supervised learning into SIRST by exploiting the consistency of paired training samples obtained from data augmentation. Unlike prevalent data augmentation techniques that often rely on standard image processing pipelines designed for visible light natural images, we introduce a novel Thermodynamics-inspired data augmentation technique tailored for infrared images. It enhances infrared images by simulating energy distribution using the thermodynamic radiation pattern of infrared imaging and employing unlabeled images as references. Additionally, to replicate spatial distortions caused by variations in angle and distance during infrared imaging, we design a non-uniform mapping in positional space. This introduces non-uniform offsets in chromaticity and position, inducing desired changes in chromaticity and target configuration. This approach substantially diversifies the training samples, enabling the network to extract more robust features. We also devise an adaptive exponentially weighted loss function to address the challenge of training collapse due to imbalanced and inaccurately labeled samples. Integrating them together, we present SemiAugIR, which delivers promising results on two widely used benchmarks, e.g., with only 1/8 of the labeled samples, it achieves over 94% performance of the state-of-the-art fully supervised learning method. The source code will be released.

## 1 Introduction

Single-frame infrared small target (SIRST) detection is a crucial component of infrared (IR) search and tracking, finding diverse applications such as maritime search and rescue, as well as agricultural yield prediction Deng et al. (2016); Teutsch & Krüger (2010). SIRST detection solely leverages spatial information from a single image, offering advantages in terms of easy deployment and real-time performance, making it highly attractive for identifying fast-moving, isolated IR small targets. While conventional SIRST methods Bai & Zhou (2010); Zhang & Peng (2019) include filtering-based, local contrast-based, and low-rank-based approaches, they often entail complex feature design and parameter tuning, yielding suboptimal detection results. In recent years, data-driven deep learning methods Ren et al. (2015); Redmon & Farhadi (2018) have been applied to SIRST detection, achieving notable progress Zhang et al. (2022a;c); Li et al. (2023). However, these methods typically enhance detection performance through fully supervised learning, relying heavily on dataset quality and quantity. This dependence is particularly problematic for small IR targets, which constitute a minuscule fraction of the image and are susceptible to labeling errors. The scarcity of IR datasets and inconsistent labeling quality pose significant challenges to advancing SIRST detection research and model generalization. Consequently, the primary challenge in this field revolves around expanding scarce IR data and mitigating the reliance on precise dataset labeling.

Semi-supervised learning Berthelot et al. (2020); Li et al. (2019); Liu et al. (2020), particularly methods employing consistency regularization, is widely employed in computer vision. These methods rely on exploiting the consistency of augmented training samples. The key challenge among them

is designing augmentation techniques that effectively extract features with robust generalization capabilities. However, it's important to note that existing data augmentation techniques Wang et al. (2023); Yang et al. (2023); Ghosh & Thiery (2020) are often not tailored for IR data. IR imaging possesses unique characteristics that make it less responsive to traditional augmentation methods, such as adding noise or other perturbations. Hence, we intend to explore semi-supervised learning for SIRST detection and customize augmentation algorithms specifically for IR imaging, taking into account its distinctive characteristics.

We begin with an in-depth exploration of the unique characteristics of IR imaging, specifically thermal radiation properties. Ideally, the uniform and constant thermal radiation from the surface of an object makes the target captured by an IR image consistent with the position of the real target, with a uniform brightness that is clearly distinguishable from the background, resulting in significant contrast. From a thermodynamic perspective, the current imaging area can be viewed as a local thermodynamic system in thermal equilibrium, exhibiting stable internal energy changes. However, in reality, various factors like non-uniform target radiation, motion blur, shooting angle interference, and background thermal noise cause spatial distortion in the acquired IR image, altering both chromaticity and position compared to the actual scene. This spatial distortion disrupts the thermodynamic equilibrium in the system. Consequently, we can establish an intuitive mapping, associating each pixel in an IR image with its pixel value (chromaticity) to each microelement in a thermodynamic system, considering its contained energy. This aids in comprehending how spatial distortions in IR images manifest within the thermodynamic system.

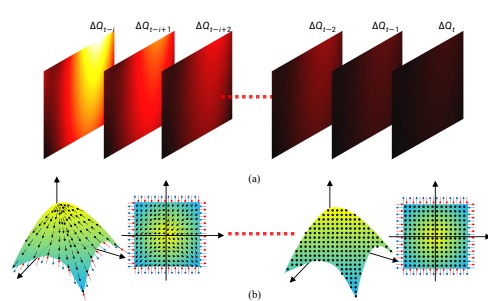

Figure 1: Schematic illustrating energy distribution changes ($\Delta Q$) in a thermodynamic system approaching a steady state. It involves changes in both macroscopic (a) and microscopic (b) energy states. The arrows in (b) indicate the direction of microelemental energy transfers.

Based on this mapping, we introduce SemiAugIR, a novel semi-supervised learning method for detecting infrared small targets. A key innovation is our thermodynamic-inspired data augmentation approach, customized for IR imaging. We leverage the thermal distribution contrast between turbulent and equilibrium states, denoted as $\Delta Q$ (Figure 1). As the thermodynamic system approaches equilibrium, heat exchange becomes directionless, and $\Delta Q$ diminishes. This allows us to create non-uniform chromaticity augmentation by aligning the temperature field function with the energy distribution changes in the IR image's microelements. To account for factors like shooting angle and motion blur affecting IR target positions, we also devise a non-uniform smoothing enhancement preserving target positions with added geometric perturbation. Furthermore, considering the severe imbalance between pixels of IR small targets and backgrounds and the variation in label quality in semi-supervised learning, we introduce a non-uniform adaptive weighted loss function. This dynamically balances simple and challenging samples based on training performance, prioritizing optimization for difficult samples, and thus addressing sample imbalance. It also prevents overfitting and handles noisy labels effectively. In summary, the main contributions of this paper are four-fold.

**1)** We present SemiAugIR, a pioneering semi-supervised approach for SIRST that significantly reduces labeling costs and improves model robustness.

**2)** We design an effective augmentation method tailored to addressing the challenges in SIRST, integrating thermodynamic simulations for infrared imaging. It enhances feature representation and discrimination in SemiAugIR by exploiting the consistency of augmented data.

**3)** We devise a novel loss function to tackle the severe target-background class imbalance issue in semi-supervised learning. It helps enhance training across networks of various sizes and datasets with varying proportions of labeled samples, leading to better performance.

**4)** Experimental results on widely-used challenging benchmarks demonstrate that our SemiAugIR delivers promising results, e.g., achieving a 94% pixel-level intersection over union (IoU) performance, with only 1/8 of labeled samples.

## 2 RELATED WORK

**SIRST Detection.** Over the past time, researchers propose a series of methods, including traditional methods (filtering-based methods Bai & Zhou (2010); Deshpande et al. (1999), local comparison-based methods Chen et al. (2013); Han et al. (2020); Hou & Zhang (2007); Han et al. (2014), low-rank-based methods Dai & Wu (2017); Gao et al. (2013); Zhang & Peng (2019); Zhang et al. (2018)) and more recent deep learning-based methods. While traditional methods require elaborate mathematical modeling and tuning of hyperparameters, deep learning-based methods Zhang et al. (2022b; 2023) learn complex nonlinear mappings in a data-driven manner that can be generalized to IR datasets of arbitrary characteristics with very promising performance. Wang *et al.* pioneer utilizing GANs Goodfellow et al. (2020) to trade off the detection rate against the false alarm rate and develop MDvsFA Wang et al. (2019). Then Dai *et al.* design an ACMNet Dai et al. (2021a) to facilitate the interaction of high-level and low-level information. Zhang *et al.* propose an ISNet Zhang et al. (2022c) to leverage image edge information using second-order Taylor finite-difference equations. Additionally, they innovate with RKFormer Zhang et al. (2022a), incorporating a transformer-based approach and optimizing the self-attention module to enhance the network's efficacy in full-feature detection. Li *et al.* present a DNANet Li et al. (2023) to facilitate the multi-scale fusion of IR small target features. The above approaches enhance IR target detection by network design, relying on full supervision where performance heavily relies on dataset quality and quantity. However, the collective amount of currently available single-frame IR data is severely limited, significantly constraining existing methods due to insufficient data for learning. Moreover, varying data labeling quality notably affects the already diminutive small IR targets. This challenge has severely impeded advancements in SIRST detection research. To overcome this bottleneck, we pioneer the application of semi-supervised learning in the domain of SIRST detection, striving to lessen dependence on labeled data and augment the existing IR dataset effectively.

**Semi-Supervised Semantic Segmentation.** The core solution to the semi-supervised problem lies in how to design a training strategy to utilize a large number of unlabeled samples to align labeled and unlabeled features to make the network more generalized. Recent semi-supervised methods typically fall into two categories: entropy minimization and consistency regularization. Entropy minimization encourages the network to output predictions with higher confidence, and the dominant approach in this regard is self-training, where training is supervised by the pseudo-labels predicted by the network. Although the accuracy of the pseudo-labels can be filtered by thresholding, they can still fall into confirmation bias. This is repeated until the network outputs low entropy predictions for unlabeled samples. Yang *et al.* develop ST++ Yang et al. (2022),showcasing that a straightforward iterative self-training approach yields good performance. Another class of methods relies on the smoothing and clustering assumptions, positing that the network generates similar predictions for identical samples with varying perturbations. Among them, Chen *et al.* propose a cross pseudo supervision strategy Chen et al. (2021) to use two independent networks to supervise each other. FixMatch Sohn et al. (2020) injects both strongly and weakly enhanced images and uses the weakly enhanced predictions to supervise the strongly enhanced ones. FlexMatch Zhang et al. (2021) and FreeMatch Wang et al. (2023) propose a course-learning method for models for different training states as well as different classes of training difficulty without introducing additional parameters. Recently, Unimatch Yang et al. (2023) demonstrates the importance of increasing the image-level perturbation space and the fact that results of consistent regularization depend on the design of sensible strong augmentations. This insight forms the theoretical basis for applying image-level data augmentation to address spatial distortion challenges in IR imaging for our SIRST detection task.

## 3 METHODOLOGY

### 3.1 NON-UNIFORM CHROMATICITY AUGMENTATION

To replicate the chromaticity change due to spatial distortion in IR images, we design a non-uniform chromaticity enhancement method based on an intuitive mapping between IR images and thermo-

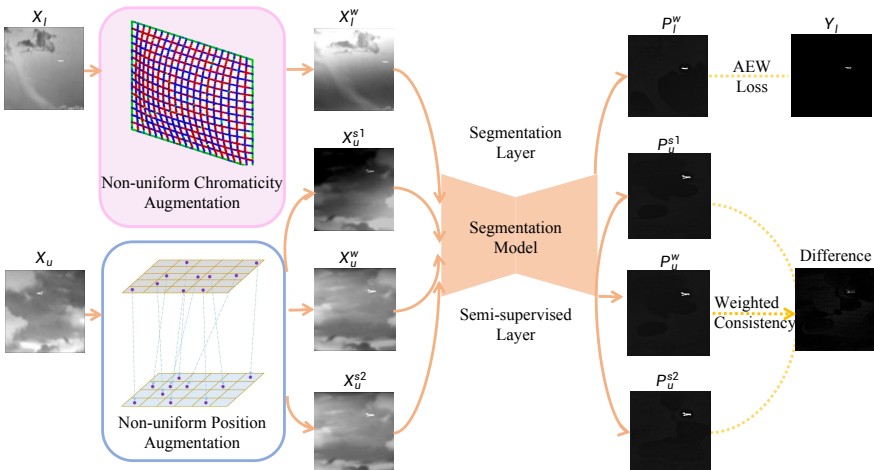

Figure 2: Overview of the proposed SemiAugIR, involving a backbone detection network, a split-head branch, and a semi-supervised branch. Both labeled and unlabeled samples undergo feature extraction via the shared backbone detection network. The split-head and semi-supervised branches process labeled and unlabeled samples to generate predictions, respectively. An AEW loss function constrains predictions from labeled samples.

dynamic systems. Our assumption is that in a thermodynamic system, the difference between the energy distribution states of the disordered and equilibrium states is denoted as $\Delta Q$. Due to the continuity of energy transfer in a thermodynamic system, the energy difference $\Delta Q$ is also continuous. After undergoing internal thermal interactions and boundary energy exchanges, the thermodynamic system tends to equilibrium and $\Delta Q$ decreases to zero. Our goal is to replicate the energy distribution of $\Delta Q$ during this process. Thus, we utilize the principle of energy continuity and non-uniformity by using a smooth two-dimensional non-uniform stochastic distribution map of chromaticity. This map simulates a random configuration of the previous energy distribution.

The procedure for generating non-uniform chromaticity augmentation is outlined herein. We aim to decompose this two-dimensional problem into two one-dimensional problems to simplify the generation of a smooth energy distribution. In the horizontal dimension, our aim is to generate five random points to conform to the cubic function $f(x)$, which serves as an energy distribution function representing the energy state of the pixel at position $x$. These five points can be regarded as a random sample on the horizontal axis. We consider the vertical thermal distribution to be highly correlated, thus assuming its smooth variation in the vertical direction. In image processing, smooth variation generally signifies that the intensity change between adjacent pixels within an image is not excessively abrupt. When we consider each pixel as a microscopic system with its energy state represented by a gray-scale value, smooth variation implies a continuous alteration in the energy state (gray-scale value) of neighboring pixels without sudden fluctuations.

To achieve this seamless transition, we can draw inspiration from the concept of a temperature field in thermodynamics. In the realm of thermodynamics, a temperature field characterizes temperature distribution at spatial points. In this context, we interpret the temperature field as a function describing the "temperature" of each point (or pixel) in an image, which significantly influences the energy state of a pixel. Within this model, we assume that variations in temperature in the vertical direction determine alterations in the energy state (i.e., the gray-scale value). In essence, if alterations in the vertical direction's temperature field value exhibit smoothness, corresponding changes in the energy state (gray-scale value) will also manifest as smooth. To implement this concept, we devise a temperature field function $T(y)$, where $y$ signifies position along the vertical axis. This function needs to satisfy two fundamental conditions: firstly, its value remains bounded across the entire image range. Secondly, its derivative (i.e., the rate of temperature change) remains bounded across the entire image range. This approach empowers us to manage alterations in the energy state along the vertical direction by adjusting function parameters to meet our specific requirements.

With these considerations, we transition the previously generated five random points horizontally into a new set of five random points in the vertical direction, smoothly transitioning them to their next positions by establishing five random endpoints. In our configuration, we employ a uniform step size for ease of implementation. When the initial random five points move to the five random endpoints, a random map featuring a smooth energy distribution is generated. It is important to note that our energy distribution maps exhibit both horizontal and vertical smooth and randomized characteristics. During the process of augmenting chromaticity in unlabeled samples, these samples can be smoothed to randomly enhance or diminish the contrast between the target and the background, simulating variations in heat.

## 3.2 NON-UNIFORM POSITION AUGMENTATION

The positions of both targets and backgrounds are prone to distortion due to inherent randomness stemming from IR imaging angles, imaging distances, and atmospheric radiative disturbances. This motivates our exploration of spatial mapping to generate spatial variations in randomness. It is worth noting that non-uniform chromaticity and positional augmentation work synergistically. Random smooth variations in chromaticity broadening the sample space of the chromaticity dimension while encouraging the network to take position information into account. Concurrently, position diversity broadens the sample space of the position dimension while encouraging the network to consider contrast information of the target and background. In practice, we represent the position $(x, y)$ of each pixel by remapping. Consider this: $g(x, y)$ is the target image, $f(x, y)$ is the source image, and $h(x, y)$ is the function of the mapping method acting on $(x, y)$:

$$g(x, y) = f(h(x, y)). \tag{1}$$

Note that the effect of the deformation enhancement depends entirely on the design of the $h(x, y)$ function. As with non-uniform chromaticity augmentation, we reduce this two-dimensional mapping problem to two one-dimensional problems to prevent excessive deformation. To have multiple stochastic effects of smooth stretching and shrinking at the same time in the same dimension, we simulate this variation using a sine function. The amplitude of the sine function is based on 60 and randomly floats up and down by 15-pixel values. The formula for $h(x, y)$ is presented as:

$$h(x, y) = a * sin(2 * \pi * t/T), \tag{2}$$

where time $T$ is randomly generated within the interval we set. We generate the target mapping by randomly taking consecutive intervals $(a, b)$ of the same size as the original image while discretizing the intervals.

## 3.3 ADAPTIVE EXPONENTIALLY WEIGHTED LOSS FUNCTION

Semi-supervised tasks often have a scarcity of labeled samples, leading to training instability. Furthermore, in the SIRST detection task, the target typically occupies a small portion of the image, causing a notable imbalance between positive and negative samples. This scenario, coupled with limited labeled data and associated noise, poses a significant challenge in training deep neural networks. Thus, we design a pioneering loss function tailored for the SIRST detection task within a semi-supervised framework. Our approach consists of designing a bounded weighted loss function that focuses on optimizing difficult samples. In addition, for binary classification in dense prediction tasks, we set optimization bounds for positive and negative samples, whereby samples beyond these bounds in prediction probability are not further optimized. Specifically, assuming the predicted output for a certain position is $p_i$, we establish weighted definitions for positive samples as:

$$loss(p_i) = \begin{cases} e^{1-p_i} \ln x & p_i < \eta \\ 0 & other. \end{cases} \tag{3}$$

We perform adaptive selection of positive and negative samples by thresholding, and the total classification loss function is:

$$L_{AEW} = \sum_{i=1}^{N} loss(p_i) / \sum_{i=1}^{N} [p_i < \eta], \tag{4}$$

where the role of $[\cdot]$ is to filter predicted values satisfyin $p_i < \eta$. Our loss function incorporates a hyperparameter $\eta$, determined based on two primary considerations. Firstly, the majority of the background regions predicted by the network lie in high-confidence intervals, with only a few difficult

samples. We can adaptively select positive and negative samples based on the network's prediction performance to prevent excessive optimization of simpler ones and better approximate confidence bounds. Secondly, manual labeling exhibits noise, particularly in boundary regions, posing difficulty in distinguishing positive from negative samples. Instead of over-optimizing high-confidence regions, we allow the network flexibility in fitting prediction boundaries to mitigate noise influence.

## 3.4 SEMI-SUPERVISED LEARNING THROUGH CHROMATICITY-POSITION CONSISTENCY

Let $D_l$ and $D_u$ denote labeled and unlabeled data, respectively, with the entire dataset represented by $D = D_l \cup D_u$. Labeled data pairs are denoted as $(x_l^i, y_l^i) \in D_l$, and unlabeled data is denoted as $x_u \in D_u$. For labeled data, we utilize our designed $L_{AEW}$ loss function for consistency supervision. The treatment for unlabeled samples is inspired by Yang et al. (2023). We define the consistency loss for soft labels as:

$$L_{con} = D(P_i, P_j), \tag{5}$$

where $D(\cdot)$ measures the similarity of two predictions. We leverage L1 loss function to measure the difference and optimize it. We perform two non-uniform chromaticity (NUC) augmentation operations and one non-uniform position (NUP) augmentation operation on input $X \in D_u$, respectively.

$$p^{NUP} = M(A^{NUP}(x^u)), p_i^{NUC} = M(A^{NUC}(A^{NUC}(x^u))), \tag{6}$$

where $A(\cdot)$ and $M(\cdot)$ represent the augmentation operation and segmentation module. $p_1^{NUC}$ and $p_2^{NUC}$ denote two distinct NUC-augmented views, while $p^{NUP}$ signifies a NUP-augmented view. Considering that NUC augmentation produces great changes to the original target, which may lead to fundamental property differences between the augmented target and the original target, while NUP augmentation has a relatively small impact on the original target, which only produces slight changes in shape and the fundamental properties of the augmented target remain unchanged. Therefore, we plan to determine optimization weights by comparing predicted results' consistency between NUC and NUP augmented views, mitigating optimization divergence:

$$L_{un} = \frac{1}{2} Con(p^{NUP}, p_1^{NUC}, p_2^{NUC}) * D(P_1^{NUC}, p_2^{NUC})$$
$$+ \frac{1}{2} Con(p^{NUP}, p_2^{NUC}, p_1^{NUC}) * D(p_1^{NUC}, P_2^{NUC}), \tag{7}$$

where $P$ is the scalar with the gradient removed. $Con(p^{NUP}, p_1^{NUC}, p_2^{NUC})$ can be described as:

$$Con(p^{NUP}, p_1^{NUC}, p_2^{NUC}) = D(p^{NUP}, p_1^{NUC})/(D(p^{NUP}, p_1^{NUC}) + D(p^{NUP}, p_2^{NUC})). \tag{8}$$

$Con(p^{NUP}, p_2^{NUC}, p_1^{NUC})$ mirrors the above formula. We refrain from assuming the absolute truth of a specific augmentation prediction, mitigating confirmation bias to some extent. The NUP-augmented prediction guides network optimization direction but isn't absolute. It helps calculate weights for NUC-augmented predictions, acting as a compromise to the rough assumption that 'NUP-augmented predictions are true.' Both NUC and NUP augmented predictions are mutually weighted, showing stable similarity after iterative optimization. The total loss $L_{all}$ can be expressed:

$$L_{all} = L_{con} + \lambda L_{un}, \tag{9}$$

$\lambda$ is a hyper-parameter set to 0.5. The consistency supervision loss $L_{con}$ is described as:

$$L_{con} = L_{AEW}(M(A^{NUP}(x_l)), y_l). \tag{10}$$

## 4 EXPERIMENT

### 4.1 EXPERIMENTAL SETTINGS.

**Dataset and evaluation metrics** We conduct extensive experiments on our SemiAugIR on the publicly available NUDT-SIRST Li et al. (2023) and NUAA-SIRST Dai et al. (2021b). NUAA-SIRST has 427 IR images with backgrounds such as clouds, cities, and oceans. NUDT-SIRST contains the largest number of background categories, including clouds, cities, oceans, fields, and bright lights. For all datasets, we employ only twenty percent for testing and eighty percent for training.

Table 1: Comparison with SOTA methods in $IoU(\%)$, $nIoU(\%)$, $P_d(\%)$, $F_a(10^{-6})$ on NUDT-SIRST and NUAA-SIRST. 'Full', 'Semi' represent fully-supervised learning and semi-supervised learning, respectively. 1/4, 1/8, 1/16, 1/32 are the ratio of labeled samples in the datasets. For NUDT-SIRST is '1/8, 1/16, 1/32', and for NUAA-SIRST is '1/4, 1/8, 1/16'.

| Methods | Description | NUDT-SIRST | | | NUAA-SIRST | | | Average | | |
|---|---|---|---|---|---|---|---|---|---|---|
| | | $IoU$ | $P_d$ | $F_a$ | $IoU$ | $P_d$ | $F_a$ | $IoU$ | $P_d$ | $F_a$ |
| Top-Hat | Filtering | 20.72 | 78.41 | 166.70 | 7.14 | 79.84 | 1012.00 | 13.93 | 79.13 | 589.35 |
| TLLCM | Local Contrast | 7.06 | 62.01 | 46.12 | 11.03 | 79.47 | 7.27 | 9.05 | 70.74 | 26.30 |
| MSPCM | Local Contrast | 5.86 | 55.87 | 115.96 | 12.38 | 83.27 | 17.77 | 9.12 | 69.57 | 66.87 |
| IPI | Low Rank | 17.76 | 74.49 | 41.23 | 25.67 | 85.55 | 11.47 | 21.72 | 80.02 | 26.35 |
| PSTNN | Low Rank | 14.85 | 66.13 | 44.17 | 22.40 | 77.95 | 29.11 | 18.63 | 72.04 | 36.64 |
| MDvsFA | CNN Full | 45.38 | 86.03 | 200.71 | 61.77 | 92.40 | 64.90 | 53.58 | 89.22 | 132.81 |
| ALCNet | CNN Full | 61.78 | 91.32 | 36.36 | 67.91 | 92.78 | 37.04 | 64.85 | 92.05 | 36.70 |
| ISNet | CNN Full | 71.27 | 96.93 | 96.84 | 72.04 | 94.68 | 42.46 | 71.66 | 95.81 | 69.65 |
| Res34-Unet | CNN Full | 90.28 | 97.99 | 5.89 | 77.18 | 96.28 | 9.23 | 83.73 | 97.14 | 7.56 |
| | CNN Full 1/32&1/16 | 46.35 | 85.16 | 160.25 | 48.45 | 90.16 | 60.27 | 47.40 | 87.66 | 110.26 |
| | CNN Full 1/16&1/8 | 57.04 | 87.17 | 79.24 | 53.58 | 92.12 | 30.14 | 55.31 | 89.65 | 54.69 |
| | CNN Full 1/8&1/4 | 76.15 | 92.19 | 47.12 | 58.94 | 93.19 | 23.19 | 67.55 | 92.69 | 35.16 |
| | ST++ 1/32&1/16 | 49.88 | 89.31 | 110.28 | 52.19 | 92.38 | 33.12 | 51.04 | 90.85 | 71.70 |
| | ST++ 1/16&1/8 | 64.91 | 92.17 | 93.29 | 58.62 | 92.81 | 21.12 | 61.77 | 92.49 | 57.21 |
| | ST++ 1/8&1/4 | 80.72 | 93.12 | 17.19 | 62.14 | 93.01 | 17.19 | 71.43 | 93.07 | 17.19 |
| | CPS 1/32 | 50.18 | 88.86 | 126.28 | 52.49 | 91.79 | 29.17 | 51.34 | 90.33 | 77.73 |
| | CPS 1/16 | 61.93 | 90.79 | 94.12 | 57.38 | 93.10 | 24.76 | 59.67 | 91.95 | 59.44 |
| | CPS 1/8&1/4 | 76.53 | 91.48 | 57.12 | 61.94 | 93.76 | 18.92 | 69.24 | 92.62 | 38.02 |
| | CPS Semi 1/32&1/16 | 54.88 | 86.93 | 191.68 | 55.19 | 92.10 | 27.19 | 55.04 | 89.52 | 109.44 |
| | CPS Semi 1/16&1/8 | 64.94 | 93.47 | 104.34 | 58.93 | 93.01 | 23.10 | 61.94 | 93.24 | 63.72 |
| | CPS Semi 1/8&1/4 | 78.53 | 95.14 | 54.82 | 62.72 | 93.12 | 18.29 | 70.63 | 94.13 | 36.56 |
| | CNN Semi 1/32&1/16 | 77.61 | 92.05 | 21.04 | 61.58 | 92.98 | 21.94 | 69.60 | 92.52 | 21.49 |
| | CNN Semi 1/16&1/8 | 82.14 | 93.47 | 9.29 | 62.48 | 93.86 | 13.01 | 72.31 | 93.67 | 11.15 |
| | CNN Semi 1/8&1/4 | 85.07 | 95.45 | 11.60 | 66.40 | 94.58 | 7.24 | 75.74 | 95.02 | 9.42 |
| ACMNet | CNN Full | 75.19 | 96.36 | 18.18 | 64.92 | 90.87 | 12.76 | 61.17 | 91.31 | 26.25 |
| | CNN Full 1/32&1/16 | 58.42 | 89.20 | 48.83 | 44.19 | 87.69 | 90.12 | 51.31 | 88.45 | 69.48 |
| | CNN Full 1/16&1/8 | 64.94 | 92.90 | 30.40 | 50.28 | 89.17 | 62.18 | 57.61 | 91.04 | 46.29 |
| | CNN Full 1/8&1/4 | 68.80 | 94.89 | 32.24 | 57.79 | 91.28 | 47.16 | 63.30 | 93.09 | 39.70 |
| | CNN Semi 1/32&1/16 | 61.04 | 91.76 | 63.57 | 48.19 | 88.13 | 76.12 | 54.62 | 89.95 | 69.85 |
| | CNN Semi 1/16&1/8 | 69.19 | 92.32 | 23.67 | 55.28 | 90.24 | 60.29 | 62.24 | 91.28 | 41.98 |
| | CNN Semi 1/8&1/4 | 73.21 | 95.74 | 23.38 | 60.29 | 92.15 | 27.19 | 66.75 | 93.95 | 25.29 |
| DNANet | CNN Full | 91.89 | 98.24 | 1.79 | 79.86 | 97.96 | 15.50 | 82.14 | 97.64 | 12.25 |
| | CNN Full 1/32&1/16 | 69.39 | 90.36 | 65.27 | 52.14 | 89.77 | 75.28 | 60.77 | 90.07 | 70.28 |
| | CNN Full 1/16&1/8 | 76.29 | 94.98 | 22.10 | 67.07 | 93.91 | 29.18 | 71.68 | 94.45 | 25.64 |
| | CNN Full 1/8&1/4 | 84.30 | 95.88 | 17.89 | 70.06 | 94.28 | 18.21 | 77.18 | 95.08 | 18.05 |
| | CNN Semi 1/32&1/16 | 76.58 | 91.19 | 53.15 | 56.79 | 89.17 | 70.12 | 66.69 | 90.18 | 61.64 |
| | CNN Semi 1/16&1/8 | 84.59 | 95.71 | 20.90 | 69.89 | 93.10 | 20.12 | 77.24 | 94.41 | 20.51 |
| | CNN Semi 1/8&1/4 | 90.12 | 96.02 | 6.85 | 73.12 | 95.12 | 10.28 | 81.62 | 95.57 | 8.57 |

**Evaluation metrics.** We utilize the pixel-level evaluation metrics intersection over union ($IOU$), probabily of detection ($P_d$), and false alarm rate ($F_a$) to measure the performance of IR detection. These three metrics are chosen because they are more sensitive to the prediction of small targets, and a few pixel-level errors may have a relatively large magnitude performance on the IOU.

**Implementation details.** To facilitate the training, we uniformly resize all the images to $256 \times 256$. Utilizing the efficient AEW loss function, we achieve SOTA performance with low complexity, employing only the Unet network with a pre-trained Resnet34 backbone. For training efficiency, we apply this network as our segmentation model. Each batch contains 4 labeled and 2 unlabeled images. The initial learning rate is set to 0.0001, employing an adamw optimizer. Parameters remain consistent for different datasets. We utilie non-uniform luminance enhancement as $A^s$. The original images are horizontally and vertically inverted with a probability of 0.2, while non-uniform position enhancement is performed with a probability of 0.6 to form $A^s$. Unlabeled samples are exclusively used in the semi-supervised branch, while labeled samples are fed into the detection branch.

Table 2: Ablation study of NUC and NUP augmentations in $IoU(\%)$, $nIoU(\%)$, $P_d(\%)$, $F_a(10^{-6})$ on NUDT-SIRST.

| Augmentation | | 1/32 | | | 1/16 | | | 1/8 | | | 1 | | | Average | | |
|---|---|---|---|---|---|---|---|---|---|---|---|---|---|---|---|---|---|
| NUCAug | NUPAug | $IoU$ | $P_d$ | $F_a$ | $IoU$ | $P_d$ | $F_a$ | $IoU$ | $P_d$ | $F_a$ | $IoU$ | $P_d$ | $F_a$ | $IoU$ | $P_d$ | $F_a$ |
| - | - | 72.28 | 93.47 | 45.03 | 76.68 | 94.31 | 35.30 | 79.61 | 96.02 | 30.29 | 87.51 | 96.88 | 10.93 | 79.02 | 95.17 | 30.39 |
| + | - | 74.37 | 93.89 | 32.10 | 79.79 | 93.89 | 20.67 | 83.25 | 95.75 | 19.87 | 89.78 | 97.36 | 6.69 | 81.80 | 95.22 | 19.83 |
| - | + | 73.83 | 94.03 | 30.52 | 78.60 | 93.47 | 18.02 | 83.00 | 95.45 | 20.27 | 89.37 | 97.18 | 6.35 | 81.20 | 95.03 | 18.79 |
| + | + | 77.61 | 94.05 | 21.94 | 82.14 | 95.47 | 9.27 | 85.07 | 96.45 | 11.60 | 90.28 | 97.99 | 5.89 | 83.78 | 95.99 | 12.18 |

Table 3: Ablation study of the proposed loss functions in $IoU(\%)$, $nIoU(\%)$, $P_d(\%)$, $F_a(10^{-6})$ on NUDT-SIRST.

| Method | Loss | 1/32 | | | 1/16 | | | 1/8 | | | 1 | | | Average | | |
|---|---|---|---|---|---|---|---|---|---|---|---|---|---|---|---|---|
| | | $IoU$ | $P_d$ | $F_a$ | $IoU$ | $P_d$ | $F_a$ | $IoU$ | $P_d$ | $F_a$ | $IoU$ | $P_d$ | $F_a$ | $IoU$ | $P_d$ | $F_a$ |
| ACMNet | IoULoss | 54.29 | 86.36 | 46.29 | 63.12 | 91.76 | 44.57 | 65.24 | 93.67 | 38.21 | \ | \ | \ | 60.88 | 90.6 | 43.02 |
| | AEWLoss | 58.42 | 89.20 | 48.83 | 64.94 | 92.90 | 30.40 | 68.8 | 94.89 | 32.24 | 75.19 | 96.36 | 18.18 | 66.84 | 93.34 | 32.41 |
| | Ours | 61.04 | 91.76 | 63.57 | 69.19 | 92.32 | 23.67 | 73.21 | 95.74 | 23.38 | \ | \ | \ | 67.81 | 93.27 | 36.87 |
| DNANet | IoULoss | 63.57 | 89.09 | 88.23 | 73.19 | 94.81 | 23.89 | 76.15 | 95.03 | 20.56 | \ | \ | \ | 70.97 | 92.98 | 44.23 |
| | AEWLoss | 69.39 | 90.36 | 65.27 | 76.29 | 94.98 | 22.10 | 84.3 | 95.88 | 17.89 | 91.89 | 98.24 | 1.79 | 80.46 | 94.87 | 35.09 |
| | Ours | 76.58 | 91.19 | 53.15 | 84.59 | 95.71 | 20.9 | 90.12 | 96.02 | 6.85 | \ | \ | \ | 83.76 | 94.31 | 26.97 |

## 4.2 QUANTITATIVE RESULTS

We conduct comparative experiments on both NUDT-SIRST Li et al. (2023) and NUAA-SIRST Dai et al. (2021b) datasets. We compare with the SOTA semi-supervised frameworks: CPS Chen et al. (2021) and ST++ Yang et al. (2022) by performing a side-by-side comparison on the baseline network ResNet34-UNet Ronneberger et al. (2015). In addition, we apply our SemiAugIR method to the SOTA CNN-based IRSTD detection methods: ACMNet Dai et al. (2021a) and DNANet Li et al. (2023). And we add six fully supervised CNN-based methods (ResNet34-UNet, MDvsFA Wang et al. (2019), ALCNet Dai et al. (2021b), ACMNet Dai et al. (2021a), DNANet Li et al. (2023), IS-Net Zhang et al. (2022c)), and five traditional methods (TopHat Bai & Zhou (2010), TLLSM Chen et al. (2013), MSPCM Moradi et al. (2018), IPI Gao et al. (2013), PSTNN Zhang & Peng (2019)).

The quantitative results are shown in Table 1. We first validate the superiority of our proposed Semi-AugIR method on the SIRST detection task on the baseline network ResNet34-UNet. Even on the dataset containing only 1/32 labeled samples, the network outperforms all the traditional methods, which indicates that our proposed semi-supervised strategy still has a far better performance than the traditional algorithms while using only a very small amount of labeling resources. Compared with the current mainstream semi-supervised frameworks CPS and ST++, our proposed method has the best performance on datasets containing different percentages of labeled samples. In addition, when our plug-and-play SemiAugIR is integrated into the CPS framework, its performance on the baseline network is greatly improved compare to CPS only, with 2% improvement in IoU and 2.3% reduction in Fa at 1/8 scale on NUDT-SIRST.

We also apply SemiAugIR to two representative fully-supervised SIRST detection algorithms, ACMNet and DNANet, and both of them can produce large detection performance gains. ACM-Net, a lightweight UNet-based detection algorithm, is highly sensitive to data quantity and labeling quality. When labeled samples are reduced or label noise is significant, ACMNet's training becomes unstable and may crash. Therefore, ACMNet serves as a valuable benchmark for assessing the robustness and training stability of our proposed method. The experimental results show that with SemiAugIR for training, ACMNet outperforms the results of fully supervised training on datasets with different proportions of labeled samples than the same proportion of data. While DNANet is currently considered the state-of-the-art (SOTA) algorithm with its densely connected network design, which enhances robustness against various disturbances, it's important to note that the available infrared (IR) data is limited in quantity. Large-scale network training on such a small dataset can lead to overfitting. Our method reliably expands the existing dataset, and the network performance can be comparable to that of the same proportion of data when trained with SemiAugIR on NUDT-SIRST dataset containing 1/8 of the labeled samples training, the network performance can be comparable to fully supervised learning results and can achieve fully supervised 98% IoU values.

### 4.3 ABLATION STUDIES

**Impact of non-uniform data augmentation:** To verify the effectiveness of our proposed non-uniform chromaticity and positions augmentation methods, we conduct a series of ablation experiments by applying them to the baseline network and present the experimental results as shown in Table 2. It is clearly observed from the table that both of our proposed plug-and-play non-uniform data augmentation methods exhibit significant improvements with respect to the baseline method. These improvements include an increase in IoU values by 2.09% and 1.55%, respectively, along with a decrease in Fa values by $12.93e^{-6}$ and $14.53e^{-6}$, respectively. The most significant performance improvements are realized when both augmentation methods are employed concurrently. When we apply both data augmentation methods to a dataset with 1/8 labeled samples for training, in terms of IoU, our method exhibits comparable performance to fully supervised methods, reaching 94%.

**Impact of the proposed loss functions:** To verify the effectiveness of the proposed AEW loss function and progressively enhanced consistent semi-supervised loss function, we summarize the specific experimental results as shown in Table 3. We clearly observe that on the dataset containing 1/32 and 1/16 labeled samples, the simultaneous adoption of these two loss functions proposed by us achieves comparable or even better performance as compared to the case of AEWLoss only, as compared to the dataset containing 1/16 and 1/8 labeled samples. Moreover, even when the network is trained using only AEWLoss, our network performance is significantly better than that of the currently dominant IoULoss and BCELoss methods. In addition, our loss function design does not over-optimize high-confidence results, and thus is able to adaptively deal with extreme sample imbalances, allowing the network to produce satisfactory detection results even when dealing with difficult samples. In addition, our method is tolerant to the number of samples, which helps to reduce the possibility of network overfitting, and is also robust to noise, thus providing a greater advantage in reducing the false alarm rate relative to other methods. These results demonstrate the effectiveness and robustness of our proposed method in dealing with the semi-supervised target detection task, which provides strong support for improving the detection performance.

### 4.4 VISUALIZATION RESULTS

We design two sets of visualization results for comparison. First, we substantiate the effectiveness of our semi-supervised strategy on the baseline network, ResNet34-UNet, as depicted in Figure 3. We can observe that the performance of the fully-supervised method decreases dramatically as the proportion of datasets containing labeled samples decreases, while our proposed SemiAugIR significantly outperforms the SOTA semi-supervised method, CPS, on datasets with all proportions of labeled samples. In addition, the integration of our method on the CPS framework can significantly improve its performance.

Next, we demonstrate that our proposed method is plug-and-play, applicable to any type of network, and significantly improves the performance, the visualization of which is shown in Figure 4. We validate the excellence of our proposed SemiAugIR on different networks with different sizes, different training stability, and containing the best current fully supervised algorithms. Our method demonstrates outstanding performance across diverse networks, showcasing its broad applicability and suggesting a novel research direction in IRSTD detection.

## 5 CONCLUSION

In this paper, we present for the first time a method for SIRST detection using semi-supervised learning. In our SemiAugIR, we design plug-and-play non-uniform chromaticity augmentation and positional augmentations, drawing inspiration from thermodynamic energy transfer principles inherent to IR imaging. These two data augmentation methods designed specifically for IR data can replicate the effect of spatial distortion on imaging, expand the scarce IR data, and improve the generalization ability of the network effectively. In addition, our loss function for semi-supervised learning effectively solves the target-background imbalance problem and eliminates the effect of variable data labeling quality on network training. Extensive experiments show that our plug-and-play IR data enhancement method can be applied to different networks, and all of them can effectively improve the robustness of each network. This research introduces innovative methodologies and perspectives to the expansive field of SIRST detection, offering promising application prospects.

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

# A APPENDIX

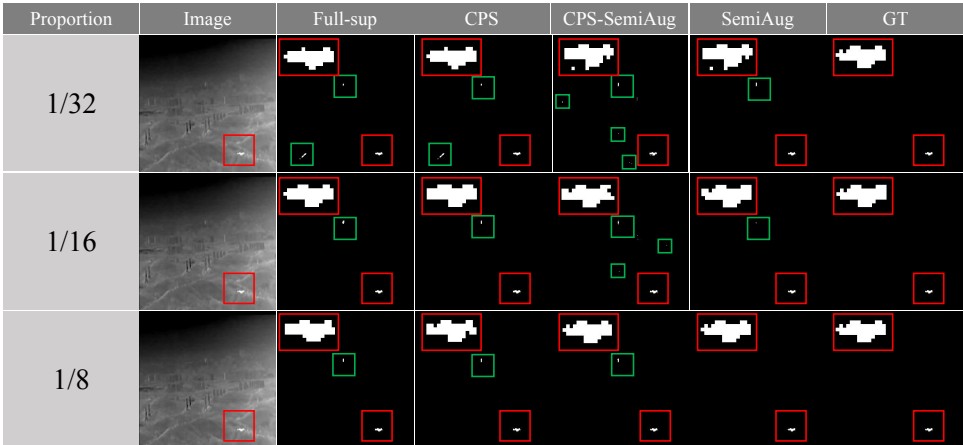

Figure 3: Visualization results of different semi-supervised learning methods over baseline network ResNet34-UNet.

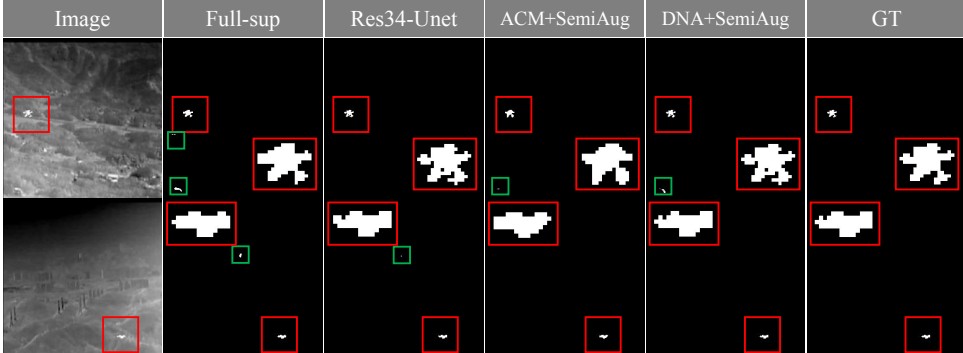

Figure 4: Visualization result of the proposed SemiAugIR over different SOTA methods.

