# OpenReview forum: "SemiAugIR: Semi-supervised Infrared Small Target Detection via Thermodynamics-Inspired Data Augmentation"
_ICLR.cc/2024/Conference — Submitted to ICLR 2024_

### Official Review · Reviewer_kGGx · 2023-10-29

**Soundness:** 4 excellent
**Presentation:** 3 good
**Contribution:** 4 excellent
**Rating:** 10
**Confidence:** 5

**Summary:**

This paper addresses the challenge of single-frame infrared small target detection (SIRST) through a semi-supervised approach names SemiAugIR, marking the pioneering instance of semi-supervised method in this domain. The author introduces a novel thermodynamics-inspired, non-uniform data augmentation technique aimed at emulating the chromaticity and positional alterations in infrared imagery caused by spatial distortions. This plug-and-play augmentation significantly amplifies the diversity of training samples, thereby enhancing the network's robustness. Additionally, the author presents an adaptive exponential loss function to effectively manage the pronounced class imbalance between targets and backgrounds. The experimental results substantiate the efficiency of the proposed method.

**Strengths:**

1. The paper is clearly written and well organized.
2. This is the first work to apply a semi-supervised method to the IRSTD task.
3. The proposed SemiAugIR can achieve over 94% performance of the SOTA fully-supervised method, while utilizing only 1/8 of the labeled samples.
4. The proposed plug-and-play non-uniform data augmentation method is well sounded and rounded, exhibits a high degree of robustness and adaptability. Its applicability extends across various infrared tasks, thereby making a valuable contribution to the advancement of infrared research.

**Weaknesses:**

1. Non-uniform chromaticity enhancement, as one of the pivotal contributions in this paper, necessitates a more comprehensive exposition. In addition to employing the translation of five key points for data generation, the author has harnessed specific techniques and threshold settings to ensure that the chromaticity enhancement results align with the intended expectations. These techniques warrant an in-depth elucidation.
2. In Section 3.4, the author conducts an analysis of the proposed Non-uniform Chromaticity Enhancement (NUC) and Non-uniform Position Enhancement (NUP), categorizing NUC as robust enhancement and NUP as mild enhancement. Is this distinction related to the concepts of strong and weak augmentation in semi-supervised learning? Furthermore, it is advisable to provide the theoretical basis for the demarcation of strong and weak augmentation to substantiate the rationale for this division.
3. The author introduces the AEWLoss as a means to address class imbalance issues; however, the paper only expounds on its treatment of positive samples. To enhance readability and the comprehensiveness of the article, please supplement the elucidation of the treatment for negative samples along with the corresponding formulas.
4. Table 2 clearly demonstrates the effectiveness of the non-uniform data augmentation method proposed in this paper. However, the author has not expounded on the foundational data augmentation methods used for comparison. It would be beneficial to include a brief description of the baseline data augmentation methods.
5. Is there any inconsistency in the magnification scale in the visual comparison figures? A meticulous examination is recommended to provide more accurate and intuitive visual contrast.

**Questions:**

1. We expect the authors to included more data augmentation methods for comparison.
2. It is advisable to provide the theoretical basis for the demarcation of strong and weak augmentation to substantiate the rationale for this division.
3. To enhance readability and the comprehensiveness of the article, please supplement the elucidation of the treatment for negative samples along with the corresponding formulas.

---

> ### Author Response · Authors · 2023-11-17
>
> We sincerely thank you for the careful and thoughtful comments. Below we address the key concerns.
>
> **1. We expect the authors to included more data augmentation methods for comparison.**
> We use common geometric augmentation methods such as random horizontal flipping, vertical flipping and random scaling and define them as the augmentation set NormGeo as shown in the table. In our semi-supervised framework, we observe that the NormGeo augmentation set significantly improves the performance metrics. In addition, we also use common luminance augmentation methods such as Gaussian blur, random contrast, etc., defining them as the augmentation set NormLum as shown in the table. We found that the NormLum augmentation set was still able to achieve significant gains. However, when we tried to use Gaussian noise enhancement (notated as GaussNoise), we noticed a slight decrease in the metrics. Therefore, we decided to remove this enhancement as it was not suitable for our IR detection task. Note that our proposed NUC (Non-Uniform Chromaticity Enhancement) and NUP (Non-Uniform Position Enhancement) are added to our setup of NormGeo and NormLum. It can be seen that even on top of all conventional enhancement methods, our two enhancement methods still significantly improve the performance. We wish to set the randomly sampled five points within the horizontal width of the image and sample them in a relatively uniform manner for a full fit over the range 0 to w-1 (image width of 256). Specifically, we set the following values: xa = 0, xb = x3/2, xe = 255, xc = xc, and xd = (xc+xe)/2. where x3 is a randomly generated value in the range 0-255. With this setup, we ensure that the horizontal coordinates of the five points are within the horizontal width of the image and are relatively evenly distributed.
> | Augmentation               | 1/32  | 1/16  | 1/8   |
> | -------------------------- | ----- | ----- | ----- |
> | NoAug                      | 58.67 | 61.25 | 64.81 |
> | NormGeo                    | 65.12 | 69.73 | 73.29 |
> | NormGeo+NormLum            | 72.73 | 77.35 | 79.63 |
> | NormGeo+NormLum+GaussNoise | 71.35 | 76.28 | 78.38 |
> | NormGeo+NormLum+NUC        | 74.37 | 79.79 | 83.25 |
> | NormGeo+NormLum+NUC+NUP    | 77.61 | 82.14 | 85.07 |
>
> **2. In addition to employing the translation of five key points for data generation, the author has harnessed specific techniques and threshold settings to ensure that the chromaticity enhancement results align with the intended expectations. These techniques warrant an in-depth elucidation.**
>
> To ensure that the five points generated are moderately and uniformly distributed over the horizontal range of the image, we carefully design five horizontal coordinates labeled x1, x2, x3, x4, and x5. We aim to arrange these five points within the horizontal width of the image and ensure a full fit over the range of 0 to w-1 (an image width of 256) by sampling them relatively uniformly. Specifically, we specify the following values: x1 = 0, x2 = x3/2, x5 = w-1, x3 = x3, and x4 = (x3+x5)/2. where x3 is a randomly generated value in the range 0-255. With this setup, we ensure that the horizontal coordinates of the five points are within the horizontal width of the image and are relatively evenly distributed. At the same time, we set the luminance values of the five points to z1, z2, z3, z4, z5. z3 is set to 0, which pulls it back to the vertical axis at z=0 to serve as a reference standard. z1 and z5 are randomly generated values, while z2 is half the value of z1 and z4 is half the value of z5. This setup makes the intensity of change in the luminance values of the five points relatively reasonable and avoids localized drastic or flat changes. In this way, we prevented the fitting results from clustering heavily around the boundaries of the maximum or minimum values of luminance, and avoided the tendency for the fitting results to be uniformly distributed or heavily disturbed. The generated results were normalized for the subsequent enhancement process. We set the maximum pixel offset to 80 and multiplied it by the previously generated normalized pixel value offset factor. The generated pixel value offsets were then added to the image and the result was truncated to the 0-255 range to generate the enhanced image. In summary, we have successfully generated the input image that has been enhanced with non-uniform luminance. The above describes the technical details of our NUC augmentation process.

---

> > ### Author Response · Authors · 2023-11-17
> >
> > **3. In Section 3.4, the author conducts an analysis of the proposed Non-uniform Chromaticity Enhancement (NUC) and Non-uniform Position Enhancement (NUP), categorizing NUC as robust enhancement and NUP as mild enhancement. Is this distinction related to the concepts of strong and weak augmentation in semi-supervised learning? Furthermore, it is advisable to provide the theoretical basis for the demarcation of strong and weak augmentation to substantiate the rationale for this division.**
> >
> > The concepts of "robust enhancement" and "mild enhancement" in this paper are related to "strong enhancement" and "weak enhancement" in semi-supervised learning. We refer to "robust enhancement" as corresponding to "strong enhancement" and "mild enhancement" as corresponding to "weak enhancement". The categorization of enhancements in this study follows the definition of a previous study [1]. That study considered color-related enhancements as "strong enhancements" because this type of enhancement randomly or regularly shifts the pixel values of the input samples in a more perturbed manner, *i.e.*, all pixels experience some degree of variation. On the contrary, methods such as geometric enhancement and random cropping are categorized as "weak enhancement", i.e., some parts of the image are not changed or the whole image is not changed, but only rotated. Therefore, these enhancements change the original image of the sample less, *i.e.*, the output of most of the pixels should be less changed compared to the output of the original image. Considering that chromaticity changes in IR images may lead to large changes in the image, and that positional enhancement is a type of geometric enhancement, we classify NUC enhancement as "robust enhancement" and NUP enhancement as "mild enhancement".
> >
> > [1] Zhao Z, Yang L, Long S, et al. Augmentation Matters: A Simple-yet-Effective Approach to Semi-supervised Semantic Segmentation[C]//Proceedings of the IEEE/CVF Conference on Computer Vision and Pattern Recognition. 2023: 11350-11359.
> >
> >
> >
> > **4. The author introduces the AEWLoss as a means to address class imbalance issues; however, the paper only expounds on its treatment of positive samples. To enhance readability and the comprehensiveness of the article, please supplement the elucidation of the treatment for negative samples along with the corresponding formulas.**
> >
> > For the screening of positive samples, we set a threshold \eta_h, and samples with scores below this threshold will be selected for model optimization. Similarly, for the screening of negative samples, we take a similar approach but need to set another threshold \eta_l, and samples with scores higher than this threshold will be included in the optimization. It is important to note that the determination of positive and negative samples is performed based on real labels. In our experiments, we chose to set \eta_l to 0.1 and \eta_h to 0.99. For pixel p_i in arbitrary position, we use the following formula to calculate the weights of negative samples as well as the loss function:
> > $$
> > w(p_i)=\begin{cases}
> > e^{p_i}, p_i>\eta_l \\\\
> > 0,\mbox{others}
> > \end{cases}
> > $$
> >
> > $$
> > Loss(p_i)=w(p_i)ln(1-p_i)
> > $$
> >
> > Similarly, the weights of the positive samples and the loss function are as follows:
> >
> > $$
> > w(p_i)=\begin{cases}
> > e^{1-p_i}, p_i<\eta_l \\\\
> > 0,\mbox{others}
> > \end{cases}
> > $$
> >
> > $$
> > Loss(p_i)=w(p_i)ln(p_i)
> > $$
> > We determine whether the corresponding position is a positive or negative sample by the true label, and then select the corresponding weights respectively, and the total classification loss function is:
> > $$
> > L_{AEW}=\sum_{i=1}^N[Loss(p_i)]/\sum_{i=1}^N[w(p_i)]
> > $$
> > Where [.] serves to select the corresponding loss(p_i) and w(p_i) based on the true labeling to determine whether p_i is a positive or negative sample.

---

> > > ### Author Response · Authors · 2023-11-17
> > >
> > > **5. Table 2 clearly demonstrates the effectiveness of the non-uniform data augmentation method proposed in this paper. However, the author has not expounded on the foundational data augmentation methods used for comparison. It would be beneficial to include a brief description of the baseline data augmentation methods.**
> > >
> > > The base data enhancement methods we use include geometric transformations such as horizontal flipping, vertical flipping, and random 90-degree rotations, as well as luminance enhancement techniques such as randomly adjusting contrast and channel panning. Our baseline model is the result obtained using these basic enhancement techniques. On top of these basic enhancements, we introduce our proposed novel enhancement methods, NUC and NUP, which further significantly improve the performance. The compatibility of our approach with generalized enhancement tricks shows that our approach is not only very effective, but can also replace traditional enhancement methods to provide significant performance enhancement.
> > >
> > >
> > >
> > > **6. Is there any inconsistency in the magnification scale in the visual comparison figures? A meticulous examination is recommended to provide more accurate and intuitive visual contrast.**
> > >
> > > We double-checked the visualization results and found that there were indeed inconsistencies in magnification as well as alignment issues, and we double-checked and realigned each rendering. The revised visuals have been updated in the Appendix.

---

### Official Review · Reviewer_5Vyd · 2023-10-30

**Soundness:** 3 good
**Presentation:** 2 fair
**Contribution:** 4 excellent
**Rating:** 5
**Confidence:** 3

**Summary:**

This paper proposes a novel semi-learning approach, image augmentation method and loss function for Infrared small target detection. These methods decline the training samples and conquer the extreme imbalance between the target and the background. Consequence, the exist networks achieve the promising performance.

**Strengths:**

The semi-supervised approach for SIRST is pioneering. The augmentation method is novel, from the perspective of thermal radiation and model the IR image augmentation by the thermodynamic system. They also devise a loss function for SIRST which conquer the imbalance issue existing pervious work, leading to better performance.

**Weaknesses:**

The methodology section of augmentation part is unclear. The overview figure is informal and ambigous.

**Questions:**

The equation(2) and (3) might be incorrect.

---

> ### Author Response · Authors · 2023-11-20
>
> We sincerely thank you for the careful and thoughtful comments. Below we address the key concerns.
>
> **The methodology section of augmentation part is unclear.**
>
> To enhance the readability of the data augmentation section, we added pseudo-code for non-uniform chromaticity augmentation operations.  Our approach is to fit a non-uniform smooth random chromaticity offset in the horizontal and vertical directions. Note that x, y represents the position of the pixel in the image, and z is the chroma value of the pixel.
>
> ```
> Algorism 1
> Input: Five randomly sampled pixels 			 A1(xa,ya,za1),B1(xb,yb,zb1),C1(xc,yc,zc1),D1(xd,yd,zd1),E1(xe,ye,ze1), and their corresponding pixels after augmentation A2(xa,ya,za2),B2(xb,yb,zb2),C2(xc,yc,zc2),D2(xd,yd,zd2),E2(xe,ye,ze2).
> Output: Chromaticity offset image set M.
> for i in range(N), N is the number of augmented image pairs
>     define Mi as the set of sets of image positions as well as luminance sets
> 	for y in range(h), h is the maximum height value of the image.
> 		for P{A,B,C,D,E}, j in (a,b,c,d,e)
> 			P(j) = (xj, zj)
> 			zj = zj1 + (zj2-zj1) * y/(h-1)
> 		Fitting the chromaticity-augmented cubic function through the set P:
>         Z(x) = ax^3 + bx^2 + cx^1 + d
>         for x in range(w), w is the maximum width value of the image.
>             z = ax^3 + bx^2 + cx^1 + d
>             add the set {(x,y), z} of 2D coordinates and chroma value to the set Mi.
>     generate chromaticity offset image Mi = {{(x,y), z},...}
>     M.add(Mi), Add Mi to the set M
> Return M
> ```
>
>
>
> **The overview figure is informal and ambigous.**
>
> We redrawed the overview figure for better comprehension.  The revised figure have been updated to the paper.
>
> **The equation(2) and (3) might be incorrect.**
>
> **Equation(2):** To avoid possible misunderstandings, we modify and elaborate on Eq. 2. To be precise, this formula describes the process of mapping the values taken, rather than the mapping function itself. Note that the coordinate points themselves are discrete, so this function is used to generate discrete offsets for interpolating the mapping. We amend the formula as follows:
> $$
> T(t)=a*sin(2*\pi*t/T)
> $$
> To avoid excessive distortion, we convert the 2D mapping to a 1D mapping. We randomly generate a discrete set X of coordinates with an interval of 1 between points in the set, for a total of 256 points. We then bring these 256 points into the function T(t) to generate the set of positional offsets O = {(x, T(x))}. Next, we randomly choose with equal probability to perform the positional offset transformation in either the horizontal or vertical direction. We first use the previously defined set X of coordinates and generate a repeating two-dimensional matrix in the horizontal or vertical direction based on the set X, depending on whether the selection is horizontal or vertical. We then identify the points in the set O of positional offsets according to the horizontal or vertical coordinate x, and repeatedly add them to the two-dimensional matrix. In this way, we generate a point-to-point position mapping. Since the mapped points are not necessarily integers, we fit these points using an interpolation function to generate pixel values for the integer coordinates of the target points. Note that there are two parameters in the formula, T and a. Where T controls the period, which is the period of change of a single axis. With this period function, we can realize the effect of simultaneous telescoping on the same axis in the image, producing relatively rich but not drastic regular changes. The parameter a is the amplitude, which is used to directly control the intensity of the change. We use 15 as the amplitude of regulation, which is a more suitable parameter obtained through experiments. This is a detailed description of the NUP formula
>
> **Equation(3):** the loss function in Equation 3 is presented only in terms of positive samples, which has certain shortcomings and lacks comprehensiveness. For the screening of positive samples, we set a threshold \eta_h, and samples with scores below this threshold will be selected for model optimization. Similarly, for the screening of negative samples, we take a similar approach but need to set another threshold \eta_l, and samples with scores higher than this threshold will be included in the optimization. It is important to note that the determination of positive and negative samples is performed based on real labels. In our experiments, we chose to set \eta_l to 0.1 and \eta_h to 0.99. For pixel p_i in arbitrary position, we use the following formula to calculate the weights of negative samples as well as the loss function:
> $$
> w(p_i)=\begin{cases}
> e^{p_i}, p_i>\eta_l \\
> 0,\mbox{others}
> \end{cases}
> $$
>
> $$
> Loss(p_i)=w(p_i)ln(1-p_i)
> $$
>
> Similarly, the weights of the positive samples and the loss function are as follows:
> $$
> w(p_i)=\begin{cases}
> e^{1-p_i}, p_i<\eta_l \\
> 0,\mbox{others}
> \end{cases}
> $$
>
> $$
> Loss(p_i)=w(p_i)ln(p_i)
> $$

---

### Official Review · Reviewer_ARSd · 2023-10-31

**Soundness:** 1 poor
**Presentation:** 2 fair
**Contribution:** 1 poor
**Rating:** 3
**Confidence:** 2

**Summary:**

The paper proposes a data augmentation technique, that was claimed to be inspired from thermodynamic properties, for detecting small objects in infrared (IR) imagery. The paper also proposes a semi-supervised training strategy and a loss function for IR detection. The method was tested on publicly available datasets for this task.

**Strengths:**

I can see the benefits of data augmentations tricks and semi-supervised approach for IR detection where data is scarce and objects of interest have higher degree of variation in appearance.

**Weaknesses:**

1. Scope: Target detection in IR images has a very limited scope, considering the ICLR audience. None of the methods presented seems to be beneficial for general vision/machine learning methods and therefore would not draw enough attention from the participants.

2. Technical novelty/soundness:

2a. Data augmentation: the core idea seems to be adding a random value drawn from a sine curve to the pixel value. I could not connect how this strategy is deduced from thermodynamic modeling (of temperature field). In addition, were there any experiments performed to check if random numbers from a normal distribution would lead to an inferior augmentation? Why the sine function is essential here?
It looks like non-uniform chromaticity augmentation is described within Section 3.2 for non-uniform position augmentation. From Table 2, it looks like they are different. If so, they should be described clearly highlighting the distinction between them.

2b. Adaptive loss  function: Section 3.3 text claims the loss function is designed to handle the positive-negative imbalance. Idont understand how loss function in Eqn 3 is addressing this imbalance. The probability p_i seems to be agnostic to label of the pixel.
Does the x in Eqn 2 denote location? If so, I dont understand why pixel location should be part of a loss function.

2c. Semi-supervised learning: There seems to be a rather complex loss function proposed in Eqn 7 to incorporate the unlabeled examples. The text does not explain the rational/theory/intuition as to why this loss function is appropriate for this problem (or any semi-supervised learning in general). Unless we understand what the loss function is doing, it is difficult to judge its merit.

3. Evaluation: The measures for evaluation needs to be explained and supported by referring to past studies that also used it. Since the measures are not the conventional ones, this is essential for the paper to clarify. Just an example, readers from natural image object detection will be confused why IoU is used for accuracy -- it is typically used to match the predictions with GT to compute mAP.

**Questions:**

..

---

> ### Author Response · Authors · 2023-11-15
>
> We sincerely thank you for the careful and thoughtful comments. Below we address the key concerns.
>
> **2a. Data Augmentation**
>
> **2a.1. The core idea seems to be adding a random value drawn from a sine curve to the pixel value.**
>
> The core idea of non-uniform chromaticity enhancement proposed in this study is different from the traditional approach of adding random values drawn from a sine curve to the pixel values (i.e., Gaussian noise enhancement). Our approach is to fit a non-uniform smooth random chromaticity offset in the horizontal and vertical directions, the implementation of which is described in pseudocode below. Note that (x, y) representing the position of the pixel in the image, and z is the chroma value of the pixel
>
> ```
> Algorism 1
> Input: Five randomly sampled pixels A1(xa,ya,za1),B1(xb,yb,zb1),C1(xc,yc,zc1),D1(xd,yd,zd1),E1(xe,ye,ze1), and their corresponding pixels after augmentation A2(xa,ya,za2),B2(xb,yb,zb2),C2(xc,yc,zc2),D2(xd,yd,zd2),E2(xe,ye,ze2).
> Output: Chromaticity offset image set M.
> for i in range(N), N is the number of augmented image pairs
>     define Mi as the set of sets of image positions as well as luminance sets
> 	for y in range(h), h is the maximum height value of the image.
> 		for P{A,B,C,D,E}, j in (a,b,c,d,e)
> 			P(j) = (xj, zj)
> 			zj = zj1 + (zj2-zj1) * y/(h-1)
> 		Fitting the chromaticity-augmented cubic function through the set P:
>         Z(x) = ax^3 + bx^2 + cx^1 + d
>         for x in range(w), w is the maximum width value of the image.
>             z = ax^3 + bx^2 + cx^1 + d
>             add the set {(x,y), z} of 2D coordinates and chroma value to the set Mi.
>     generate chromaticity offset image Mi = {{(x,y), z},...}
>     M.add(Mi), Add Mi to the set M
> Return M
> ```
>
> **2a.2. Were there any experiments performed to check if random numbers from a normal distribution would lead to an inferior augmentation?**
>
> The sine curves mentioned in this paper are used for non-uniform positional enhancement, which are two unrelated concepts to Gaussian noise enhancement. In addition, we conducted experiments with Gaussian noise enhancement to verify that normally distributed random numbers lead to poorer enhancement, the results of the experiments are shown in Table A. We observed a slight decrease in the IoU evaluation metrics after using the Gaussian noise enhancement method for the baseline network (i.e., without NUC and NUP enhancement). This suggests to us that the enhancement method may not be sufficiently adapted to the characteristics of the infrared image, resulting in less than expected detection results. We infer that one of the main reasons for this is that IR images are grayscale maps, and thus the target has relatively little texture information, i.e., the reference information is relatively limited. When random noise is introduced, a large number of generated noise points with high chromaticity values may be confused with small IR targets, which in turn significantly disturbs the already limited detection information, thus further increasing the detection difficulty of IR images.
>
> Table A:
>
> | Augementation  | 1/32  | 1/16  | 1/8   |
> | -------------- | ----- | ----- | ----- |
> | NoAug baseline | 72.73 | 77.35 | 79.63 |
> | Gaussian noise | 71.35 | 76.28 | 78.38 |
>
> **2a.3. It looks like non-uniform chromaticity augmentation is described within Section 3.2 for non-uniform position augmentation. From Table 2, it looks like they are different. If so, they should be described clearly highlighting the distinction between them.**
>
> We propose non-uniform chromaticity enhancement and non-uniform positional enhancement, which, although they both belong to the category of non-uniform data enhancement, are two completely different kinds of enhancement aimed at achieving different goals. The differences between them are:
>
> 1. non-uniform chromaticity enhancement focuses on adjusting the variation of chromaticity, while non-uniform positional enhancement achieves the effect of non-uniform deformation on the whole image by designing the mapping function on the position.
> 2. Non-uniform chromaticity enhancement can be categorized as a type of chromaticity perturbation enhancement, while non-uniform positional enhancement belongs to the category of geometric enhancement.
> 3. The generation process of these two enhancement methods is completely different. Non-uniform chromaticity enhancement is achieved by randomly selecting points and fitting a global randomly smoothed luminance offset, while non-uniform positional enhancement directly sets the mapping function in the horizontal or vertical direction to generate the positional mapping map.
> 4. The results of the ablation experiments show that the effects of these two enhancement methods can be superimposed on each other, which side-steps the fact that they have complementary roles and each plays a different role.

---

> ### Author Response · Authors · 2023-11-15
>
> **2b. Adaptive Loss Function**
>
> **2b.1. Section 3.3 text claims the loss function is designed to handle the positive-negative imbalance.**
>
> For SIRST, the positive and negative sample imbalance problem refers to a situation of extreme imbalance between positive samples (target pixels) and negative samples (background pixels) in an infrared image. Generally, a situation where the ratio of positive and negative samples exceeds 4:1 is regarded as a positive-negative imbalance. However, in infrared image processing, taking a 256x256 infrared image as an example, the target usually occupies only a few tens of pixels (only about one ten-thousandth of the whole image), which leads to a very serious imbalance between positive and negative samples. Therefore, in order to effectively deal with this problem, a suitable loss function needs to be designed to deal with the significant imbalance between positive and negative samples.
>
> **2b.2.  Idont understand how loss function in Eqn 3 is addressing this imbalance.**
>
> The loss function in Equation 3 is presented only in terms of positive samples, which has certain shortcomings and lacks comprehensiveness. The following is an exhaustive description of the loss function we designed. Our loss function assigns greater weights to the difficult samples, as well as by setting thresholds for positive and negative samples in order to achieve adaptive sample selection optimization.
>
> For the screening of positive samples, we set a threshold \eta_h, and samples with scores below this threshold will be selected for model optimization. Similarly, for the screening of negative samples, we take a similar approach but need to set another threshold \eta_l, and samples with scores higher than this threshold will be included in the optimization. It is important to note that the determination of positive and negative samples is performed based on real labels. In our experiments, we chose to set \eta_l to 0.1 and \eta_h to 0.99. For pixel p_i in arbitrary position, we use the following formula to calculate the weights of negative samples as well as the loss function:
> $$
> w(p_i)=\begin{cases}
> e^{p_i}, p_i>\eta_l \\\\
> 0,\mbox{others}
> \end{cases}
> $$
>
> $$
> Loss(p_i)=w(p_i)ln(1-p_i)
> $$
>
> Similarly, the weights of the positive samples and the loss function are as follows:
> $$
> w(p_i)=\begin{cases}
> e^{1-p_i}, p_i<\eta_l \\\\
> 0,\mbox{others}
> \end{cases}
> $$
>
> $$
> Loss(p_i)=w(p_i)ln(p_i)
> $$
>
> We determine whether the corresponding position is a positive or negative sample by the true label, and then select the corresponding weights respectively. By w(p_i) the weights of easily categorized samples are decreased, while the weights of hard-to-categorize samples are increased. The total classification loss function is:
> $$
> L_{AEW}=\sum_{i=1}^N[Loss(p_i)]/\sum_{i=1}^N[w(p_i)]
> $$
> Where [.] serves to select the corresponding loss(p_i) and w(p_i) based on the true labeling to determine whether p_i is a positive or negative sample.
>
> **2b.3. The probability p_i seems to be agnostic to label of the pixel. Does the x in Eqn 2 denote location? If so, I dont understand why pixel location should be part of a loss function.**
>
> Our definition of the symbol x is relatively independent in each section. In Section 3.1, we introduce the symbols x and y, which represent the horizontal and vertical coordinate values in an image, respectively. These symbols are used to describe the image coordinate system. In Section 3.2, we again use the symbols x and y, which have similar meanings as in Section 3.1, to represent the horizontal and vertical coordinate values in an image. However, in Section 3.3, we need to correct the symbol x in Equation 3 in Section 3.3, which should actually be p_i. In this Section, the symbol p_i represents the predicted probability value of a point in the image, which is different from the definition of the symbols in the previous two sections. This correction is intended to accurately describe the different meanings of the symbol x in different chapters and to ensure consistency and accuracy of the notation.

---

> ### Author Response · Authors · 2023-11-15
>
> **2c. Semi-supervised Learning**
>
> **2c. There seems to be a rather complex loss function proposed in Eqn 7 to incorporate the unlabeled examples. The text does not explain the rational/theory/intuition as to why this loss function is appropriate for this problem (or any semi-supervised learning in general). Unless we understand what the loss function is doing, it is difficult to judge its merit.**
>
> The loss function proposed in Equation 7 aims to achieve consistent trade-off optimization between strong and weak enhancement. In this context, NUC enhancement is used to enhance the chromaticity of the target, while NUP enhancement is mainly used to improve the position of the target, which belong to strong and weak enhancement, respectively. Previous semi-supervised methods usually use the results of weak enhancement as a benchmark to guide the network to optimize the prediction results of strong enhancement to converge to it gradually. However, weakly-enhanced predictions may have some degree of error, so simply assuming that weakly-enhanced predictions are accurate may lead to training data bias. To address this issue, we redesigned the improved consistency loss to guide the network to optimize without absolutely assuming that a particular augmented prediction is correct. This helps to avoid the confirmation bias problem to a certain extent. The role of NUP-enhanced prediction is to guide the direction of network optimization, but it is not absolute. The NUP-enhanced prediction is used to compute the weight of the NUC-enhanced prediction in the optimization, and therefore compromises the general assumption that "the NUP-enhanced prediction is true". The NUC- and NUP-enhanced predictions are weighted against each other, and after iterative optimization, they will show a stable similarity.
>
> **3. Evaluation**
>
> **3. The measures for evaluation needs to be explained and supported by referring to past studies that also used it. Since the measures are not the conventional ones, this is essential for the paper to clarify. Just an example, readers from natural image object detection will be confused why IoU is used for accuracy -- it is typically used to match the predictions with GT to compute mAP.**
>
> Infrared small target detection essentially employs a semantic segmentation approach to achieve target detection, which is able to more accurately detect the presence of a target and retain accurate edge information without relying on the traditional anchor frame approach. In this study, we adopt several evaluation metrics commonly used in the field of infrared small target detection, including IoU, Pd, and Fa, which are all measured in pixels and widely used in the field of infrared small target detection. Their mathematical representations are shown below.
> $$
> IoU = \frac{A_{i}}{A_{u}}
> $$
> where A_i and A_u denote the intersecting and union area, respectively.
> $$
> {P_{d}} = \frac{{N_{pred}}}{{N_{all}}}
> $$
> where N_pred and N_all denote the number of correctly predicted pixels and all pixels of the target, respectively.
> $$
> {F_{a}} = \frac{{N_{false}}}{{N_{all}}}
> $$
> where N_false and N_all are pixels that are mistakenly detected and all pixels in the image, respectively.

---

### Meta-Review · Area_Chair_P4BQ · 2023-12-10

**Metareview:**

This paper proposes an thermodynamics-inspired data augmentation for achieving better infrared small target detection with semi-supervised learning.  Their augmentation includes non-uniform offsets in both chromaticity and position, substantially diversifying training samples.  Along with techniques addressing imbalanced and inaccurately labeled samples, their specialized semi-supervised method is able to deliver 94% of fully supervised learning performance with 1/8 labeled data.

The strengths of the paper are a physics inspired data augmentation method tailored for IR data and strong empirical validation in the IR domain.   The weaknesses of the paper are a topic of narrow interests and lack of general technical novelty.  The demonstrated performance gain of semi-supervised learning over supervised learning has long been seen on ImageNet, e.g. SimCLRv2 achieves 93% of fully supervised performance with 1% labeled data.

The paper has received 3 divergent reviews, to which authors have responded with detailed further clarification and additional experimental results.  The final ratings are 3/5/10, which also reflect confusing writing and limited interests for a general machine learning audience.

Weighing pros and cons of this paper and the overall luck-warm reception, the AC recommends rejection.

**Justification For Why Not Higher Score:**

A topic of narrow interests and lack of general technical novelty.

**Justification For Why Not Lower Score:**

N/A

---

### Decision · Program_Chairs · 2024-01-16

Reject